# Data-Driven Radiogenomic Approach for Deciphering Molecular Mechanisms Underlying Imaging Phenotypes in Lung Adenocarcinoma: A Pilot Study

**DOI:** 10.3390/ijms24054947

**Published:** 2023-03-03

**Authors:** Sarah Fischer, Nicolas Spath, Mohamed Hamed

**Affiliations:** 1Institute for Biostatistics and Informatics in Medicine and Ageing Research, Rostock University Medical Center, Ernst-Heydemannstr. 8, 18057 Rostock, Germany; 2Department of Systems Biology and Bioinformatics, University of Rostock, Ulmenstr. 69, 18057 Rostock, Germany; 3Department of Medicine II, Hematology and Oncology, University Hospital Schleswig-Holstein, Arnold-Hellerstr. 3, 24105 Kiel, Germany

**Keywords:** lung cancer, radiogenomics, data integration, imaging genomics

## Abstract

The heterogeneity of lung tumor nodules is reflected in their phenotypic characteristics in radiological images. The radiogenomics field employs quantitative image features combined with transcriptome expression levels to understand tumor heterogeneity molecularly. Due to the different data acquisition techniques for imaging traits and genomic data, establishing meaningful connections poses a challenge. We analyzed 86 image features describing tumor characteristics (such as shape and texture) with the underlying transcriptome and post-transcriptome profiles of 22 lung cancer patients (median age 67.5 years, from 42 to 80 years) to unravel the molecular mechanisms behind tumor phenotypes. As a result, we were able to construct a radiogenomic association map (RAM) linking tumor morphology, shape, texture, and size with gene and miRNA signatures, as well as biological correlates of GO terms and pathways. These indicated possible dependencies between gene and miRNA expression and the evaluated image phenotypes. In particular, the gene ontology processes “regulation of signaling” and “cellular response to organic substance” were shown to be reflected in CT image phenotypes, exhibiting a distinct radiomic signature. Moreover, the gene regulatory networks involving the TFs *TAL1*, *EZH2*, and *TGFBR2* could reflect how the texture of lung tumors is potentially formed. The combined visualization of transcriptomic and image features suggests that radiogenomic approaches could identify potential image biomarkers for underlying genetic variation, allowing a broader view of the heterogeneity of the tumors. Finally, the proposed methodology could also be adapted to other cancer types to expand our knowledge of the mechanistic interpretability of tumor phenotypes.

## 1. Introduction

Lung cancer is one of the most predominant cancer types that are diagnosed with a high incidence (14.3% of total male and 21.5% of female new cancer cases) and with a high mortality rate worldwide [1]. Currently, the diagnosis, prognosis, and treatment selection of lung cancer are mainly accomplished by histologic inspection of tumor tissue [2], lymph node involvement [3], radiological imaging [4], and mutational status of *EGFR, KRAS, ALK, BRAF, ROS1, HER2, RET, MET*, and *PD-L1* expression analysis [5]. Major challenges include the genetic, temporal, and spatial heterogeneity of tumors, the invasive collection of tumor samples, and the inability to distinguish between clinically relevant subtypes [6]. 

Genome-wide characterization has recently been utilized in the clinical assessment of lung cancer with multiple molecular assays, including gene expression alterations [7], miRNA expression profiles [8], and epigenetic modifications [9] such as DNA methylation status. However, these genomic sequencing assays fall short of capturing the spatial and temporal heterogeneity of tumors [10]. Medical imaging modalities such as MRI and CT have great potential to provide comprehensive details about tumor shape, intensity, and texture. Using this information as a prognostic biomarker for overall survival has already been proposed by generating a risk score from CT image features in lung cancer [11]. Furthermore, radiological imaging is used as an ongoing clinical routine to monitor tumor progression, angiogenesis, and distant metastasis to other organs [6]. There are several well-performing machine learning-based radiomic signatures for predicting *EGFR* and *KRAS* mutation status [12,13].

There is an ongoing effort to describe the biological representation of radiomic features [14]. The recent technological revolutions in clinical imaging (radiology/radiomics) and genomic technologies have led to the emergence of a new research area called “molecular imaging”, “imaging genomics”, or radiogenomics. This field refers to the study of the association between the molecular properties of tumors and their imaging phenotypes. For instance, many radiogenomics studies have reported significant correlations of molecular markers and clinical variables based on CT or MRI image features of lung [15,16,17], prostate [18], and breast neoplasms [19]. These studies hypothesized that alterations in gene expression patterns could lead to specific tumor architectures captured by non-invasive imaging. Recently, the field has gradually broadened. For example, machine and deep learning approaches have predicted mutation status based on the image features of lung tumors [13,20,21]. To improve these studies, radiomic features need to be robust to changes in the setting, such as CT or MRI scanner variables and reconstruction algorithms. Recently, a major step has been taken to define and validate the robustness of the features [22].

In contrast to unconnected molecular or imaging analyses, radiogenomics specifically outlines links between different datasets across a range of spatial and temporal scales [23]. Radiogenomic association maps (RAMs) can represent the correlation of radiomic features, genomic features, and clinical data in visually appealing graphs that reveal complex patterns [24]. Thus, the construction of RAMs could contribute to a better understanding of the tumor biology underlying imaging phenotypes and provide new insights into the identification of non-invasive surrogate biomarkers that accurately predict tumor molecular characteristics and suggest potential therapeutic approaches. This could provide an extension to the currently available methods, such as machine learning-based approaches [11,13,18]. When various molecular assays (multi-omics data) are available, RAM generation could provide more comprehensive insights than just analyzing correlations between, for example, image features and gene expression. For instance, we can learn more comprehensively how biological processes and signaling pathways are reflected in image features. Our methods for constructing RAMs consist of unsupervised cluster-based feature selection, which is well understood and has been applied to other applications such as early diabetes detection [25].

## 2. Results

### 2.1. Overview of the Radiogenomic Approach

We developed and applied a bioinformatics workflow to perform an integrative analysis of gene (mRNA) expression, miRNA expression, and clinical and imaging data (Figure 1). All patients with primary tumors were included. The common cohort size used for the integrative analysis is 22 patients with a median age of 67.5 years (min–max, 42–80) (Table 1). Image processing starts with the manual segmentation of the tumor region of interest (ROI) from patient CT scans (*n* = 69). Fiji [26] and MATLAB were used to extract and store 86 image features related to four imaging phenotypes: tumor size, texture, morphology, and shape. The expression data of all available mRNA (*n* = 515) and miRNA (*n* = 513) samples were analyzed by differential expression analysis. The resulting differentially expressed genes (DEGs) and miRNAs (DEMs) were further used to identify over-represented gene ontology (GO) functional terms using gProfiler [27]. We performed gene set enrichment analysis using GO terms and extracted image features via Piano [28]. This provides a summary statistic of the correlations of the extracted image features with the enriched GO terms. We extracted the intersection of enriched GO terms between gene and miRNA expression datasets. For these intersecting GO terms, patients were clustered into phenotypically distinct subgroups according to their gene and miRNA expression signatures using hierarchical clustering, reflecting the biological correlations of these signatures with the corresponding image features. Clinical and mutation data were added to these clusters using the ComplexHeatmap package [29] resulting in a radiogenomic association map. Finally, TFmiR2 [30] was used to construct the gene regulatory network (GRN) of these GO terms that potentially explain the phenotypic differences between patient subgroups.

### 2.2. Differential Expression and Gene Set Enrichment Analysis

Differential expression analysis yielded 7214 and 147 differentially expressed genes (DEGs) and differentially expressed miRNAs (DEMs), respectively. The postulated functional roles of these dysregulated genes and miRNAs were summarized in 317 significant GO terms (biological processes) for the DEGs and 538 terms for the DEMs (Appendix A).

Gene set enrichment analysis was performed to investigate the association between the transcriptional signatures of these significant terms and the tumor radiomic phenotypes. This revealed 7634 and 1156 significant associations between any image feature and any revealed enriched GO term for the DEGs (Appendix A) and the DEMs (Appendix A), respectively. Biological processes highly associated with radiomic phenotypes included nuclear division, cell cycle, cytokine-mediated signaling, and interleukin-6 signaling.

Only 11 GO terms overlapped between the association results of both DEGs and DEMs with the radiomic features (Figure 2A). Most of these 11 GO terms were biological processes specific to cell differentiation, such as cell population proliferation or positive regulation of developmental processes.

Interestingly, the four studied tumor phenotypes (morphology, shape, texture, and size) show different association patterns with the dysregulated genes (DEGs) and miRNAs (DEMs). For instance, most image features related to tumor size and morphology are mainly associated with DEGs but not with DEMs. Additionally, several texture features calculated based on the neighborhood gray-tone difference matrix negatively correlate with DEGs. By contrast, the texture features calculated from the gray-level run-length matrix positively correlate with DEMs.

The gene/miRNA expression values of these 11 GO terms were hierarchically clustered to form two patient clusters. We measured the similarity of the clusters between the DEG and DEM datasets by the intersection of common patients (Table 2). We also integrated patient clinical information such as age range, smoking status, tumor stage (T and N), and the incidence of the most common mutations in lung cancer: *ALK*, *EGFR*, *KRAS,* and *TP53*.

As we were interested in basic cellular and biological processes, we narrowed our analysis to the following two GO processes: (1) regulation of signaling, which shows the highest overlap of patients (17/22 = 77%) between the two patient clusters of DEGs and DEMs, and (2) cellular response to organic substances, which has the highest number of associations between the transcriptomic and image features (see Figure 2B and Table 2). We then analyze these GO terms in more detail to unravel how they were reflected in the radiomic phenotypes. The RAMs of the remaining 11 detected GO terms are depicted in more detail in Appendix A.

### 2.3. Regulation of Signaling

The regulation of the signaling process incorporates basic signaling genes and miRNAs. Previous studies have shown heterogeneous tumors exhibit different signaling mechanisms and dysregulation patterns of related genes and miRNAs [31]. Our results not only showed that regulation of signaling was significantly associated with varying tumor phenotypes but also allowed for patient clustering based on the expression signatures of the signaling genes/miRNAs (Figure 3) with significant differences in tumor morphology (tumor variance). Moreover, signaling genes and miRNAs positively correlated with tumor variance appear to have an inflammatory function, such as *hsa-mir-9* (Appendix A). Consistent with our findings, this miRNA has already been experimentally proposed as a prognostic biomarker based on its correlation with poor overall outcomes [32]. It is also noteworthy that most clinical data, such as tumor stage and mutation status, did not show significant differences between the two patient groups.

Furthermore, the DRFs calculated as a differentiator for the two groups with their fold change and *p*-values are displayed. The assigned image phenotype refers to the group of the image feature (Appendix A).

### 2.4. Cellular Response to Organic Substance

Biological processes related to response to organic substances had the highest number of significant associations between the transcriptomic and image features in lung carcinoma. This is consistent with the fact that one of the main causes of lung cancer is tobacco smoking, which contains carcinogenic substances, such as organic cyclic compounds [33], that damage lung tissue. Most of the texture features were significantly associated with the expression patterns of miRNAs and genes, with a positive correlation observed for the miRNA signature and a negative correlation for the gene signature (Figure 3). Moreover, when clustering patients based on the miRNA expression signature of the biological process “cellular response to organic substances”, patient groups tend to have significant differences in tumor texture features such as homogeneity, contrast, and coarseness (Figure 4C, Appendix A). This highlights the critical role of miRNAs in tumor texture heterogeneity in CT images of lung cancer patients exposed to organic substances. Unexpectedly, the clustering of patients based on gene expression signatures of the BP “cellular response to organic substances” revealed only morphology (i.e., variance) as a difference between the patient subgroups. Figure 4B depicts exemplary CT images for the two patient groups.

Notably and in concordance with tumor heterogeneity, inflammatory activity and previous exposure to organic cyclic compounds are positively correlated overall. Similar to the “regulation of signaling” BP, there were no clear, coherent patterns in tumor stage, mutation status, or smoking status (Figure 4A) between the patient subgroups.

Furthermore, the DRFs calculated as a differentiator for the two groups are shown with their fold changes and corresponding *p*-values. The assigned image phenotype refers to the group of the image feature (Appendix A). In contrast to one DRF (variance) of the mRNA expression-based RAM, the two groups in the miRNA RAM can be differentiated by a set of 14 image features, all belonging to the texture phenotype.

### 2.5. Regulatory Interactions Underlying Phenotypic Differences

For each of the two examined biological processes, we constructed a TF–miRNA-mediated regulatory network that combines transcriptional and post-transcriptional interactions between the associated DEGs and DEMs, potentially driving the phenotypic differences between the patient subgroups (Figure 5). The constructed networks encompass three types of molecular interactions: (1) TF → target gene, (2) miRNA → target gene, and (3) TF → miRNA, describing how miRNAs are significantly involved in controlling tumor phenotypes. For the “regulation of signaling”, we identified two main hub genes: *TAL1* and *TGFBR2,* which contribute largely to the regulation of the network (Figure 5A).

By contrast, *TGFBR2* was identified as the main hub gene for the “cellular response to organic substances” term (Figure 5B). Our results show that *TAL1* is a lung-specific gene associated with lung carcinoma and directly regulates *TGFBR2*, which was previously annotated as a tumor suppressor gene [34]. *TAL1* is also known to control normal myeloid differentiation and is an experimental drug target for the treatment of T-cell acute lymphoblastic leukemia [35]. Our analysis suggests a regulatory role for *TAL1* in controlling tumor morphology, particularly tumor variance (Figure 3C). Many studies have reported the suppressive function of *TGFBR2* in tumorigenesis [35,36], but no previous report has been able to highlight its regulatory role in governing the tumor texture and morphology (Figure 4C).

## 3. Discussion

Radiogenomic approaches combine radiological images with underlying molecular information to reveal possible links between these tumor phenotypes and the underlying biology [31]. Biologically plausible associations between gene expression, miRNA expression, and image features could have a clinical context, such as early prediction of appropriate treatments, and a positive impact on overall survival.

The decision to utilize the whole transcriptome, in addition to high-evidence genotypes like EGFR mutations, was made to include as yet unknown dysregulated genes. In addition, we did not want to reduce the already small sample size by including only a subset of the patients. For example, EGFR mutations have an estimated prevalence of only 10–16% in Caucasians and ALK adds up to 1–10% [37].

We proposed a data-driven approach to construct radiogenomic association maps (RAMs) that link imaging phenotypes to associated molecular features. These RAMs have the potential to identify image features that reflect the transcriptomic and post-transcriptomic regulations behind tumor pathogenesis. Such candidate image features could be used as surrogate biomarkers in the absence of genomic information and as an indicator of the underlying biological processes and pathways. Yeh et al. [31] applied a similar approach in breast cancer patients and found positive and negative associations between image phenotypes, such as size and KEGG pathways. In addition to the RAM-based approach, several other methods detect relationships between the image features and genetics, for example, by using PET rather than CT images and associating image features with oncogenic signaling pathways [38]. Other approaches use different methods to associate the imaging phenotypes with genetic signatures, so-called metagenes, using a correlation-based approach [17].

In addition, our approach helped to decipher the complex regulatory interactions between associated genes and miRNAs, explaining the differences between patients in tumor imaging phenotypes.

Our approach highlighted biologically plausible associations between imaging phenotypes, dysregulated genes, and miRNAs in lung tumor patients. For instance, the tumor size and morphology phenotypes were exclusively associated with gene expression profiles, whereas the texture phenotypes were associated with gene and miRNA profiles. This relationship sheds light on quantifying the regulatory role of genes and miRNAs in shaping the observed tumor phenotypes in radiological images. 

Missing interpretability of image features for clinical associations beyond the subcategories defined by image features such as shape or density complicates their evaluation. As gene ontology databases provide curated molecular knowledge, this direct connection to previous findings enables the detection of surrogate image features for biological processes involved in tumor phenotypes. Additionally, our approach visually represents the patient’s clinical and mutation data to the constructed RAM in a complex heatmap. Although no differences in clinical and mutational data of *EGFR*, *ALK*, *TP53,* and *KRAS* were observed, an equivalent analysis with a larger patient cohort could determine yet unknown patterns.

Interestingly, the genes involved in the regulation of cell signaling were found to be positively associated with shape and size image features. This connection seems biologically plausible as upregulated signaling pathways in tumors would induce proliferation and, thus, growth. Both genes and miRNAs involved in this biological process were negatively associated with tumor *variance*. This might lead to the conclusion that rapidly growing tumors lose their grayscale *variance*. Moreover, our RAM analysis shows that this image feature can be used to distinguish the signaling activity of a patient’s tumor. For instance, the miRNAs *hsa-mir-9-1, hsa-mir-9-2, and hsa-mir-9-3* are known to cause inflammation and positively correlate with tumor *variance* in patient group 1 (Figure 3, blue samples). Recent unpublished work analyzed the expression differences of several miRNAs (including *mir-9*) and showed that these miRNAs show different expression patterns in early, middle, and late tumor stages [39]. In patient group 2, the gene *DEPTOR*, which is known to inhibit lung tumorigenesis [40], is negatively correlated with tumor *variance* (Appendix A, red samples), suggesting its potential role as a diagnostic biomarker for differentiating patients at high risk of progression.

The dysregulated genes and miRNAs related to organic substances were able to distinguish patients with significant differences in tumor texture phenotype.

Of particular interest is the state of the inflammatory microenvironment of the tumor. Our results demonstrated evidence that inflammatory activity due to organic cyclic compounds (smoking) correlates with tumor texture and suggests the miRNAs hsa-mir-196a, hsa-mir-187, hsa-mir-133a, and hsa-mir-1 as a potential factor for tumor heterogeneity between patient groups. 

When constructing the gene–miRNA regulatory networks associated with the two GO terms examined, *TAL1* and *TGFBR2* were identified as hotspot genes potentially regulating these two GO terms. The stimulation of *TGFBR2* by *TAL1,* specifically in lung tissue, has not been experimentally confirmed. Lo Sardo et al. [34] described *EZH2* as a suppressor of *TGFBR2*, resulting in tumor growth mediated by a cluster of miRNAs (*miR-25*, *93,* and *106b*). Although this mechanism was not reflected in our GRN, we discovered another cluster of miRNAs (*hsa-mir-19a*, *20a*, and *21*) that may be involved in tumor growth and progression, in addition to the findings described by Lo Sardo et al. [34]. It is also noteworthy that the transcription of *ADRB2*, a target gene in the constructed regulatory network, is enhanced by the *visualized TAL1-EZH2 axis*. It is the encoding gene for beta-adrenoreceptors. In the literature, *ADRB2* has been controversially reported to be associated with proliferation, angiogenesis, tumor progression, distant metastasis, and TKI resistance [41].

### 3.1. Study Limitations

Missing freely available repositories for patients’ multi-omics data was the main challenge for this study. We thus used all matched samples to create the RAM. Therefore, the results presented in this study require larger patient cohorts with various radiogenomics profiles to validate the detected RAMs. Furthermore, many radiogenomic studies can be improved by marking the specific biopsy site in the radiomic images to correlate the tissue-specific expression with the corresponding ROI in the image.

Another important limitation is the technical challenges in data acquisition and processing, such as image standardization problems when using different CT scanners with varying parameters such as slice thickness, reconstruction algorithms, and radiation detector resolution. Finally, an automated ROI segmentation would compensate for the human bias introduced by manual segmentation. 

### 3.2. A Word of Caution

We must stress the obvious but often missed fact that association never implies causation when using RAM models. Nevertheless, we spotted literature-confirmed RAM examples generated from different OMICs datasets. Future research is warranted to test/assess the robustness and consistency of the proposed RAM map via receiver operator characteristic curves and cross-validation (CV) techniques—for instance, by building machine learning models to predict the radiographic features from the molecular data and vice versa. A second standard method to validate the detected RAMs is to apply our approach to independent/external patient cohorts and compare the identified association patterns.

## 4. Materials and Methods

### 4.1. Datasets Origin

Clinical data, and gene and miRNA expression profiles for lung adenocarcinoma patients were downloaded from The Cancer Genome Atlas (TCGA) portal, namely the TCGA-LUAD project [42]. Genomic datasets were collected at level three. The matching CT studies (imaging traits) were obtained from The Cancer Imaging Archive (TCIA) [43] (Appendix A).

### 4.2. Image Data Analysis

The DICOM images were loaded as image sequences into the ImageJ2 software [44] and segmented using the segmentation manager plugin of Fiji V.8 [26] to create the regions of interest (3D ROIs) delineating the tumor in each CT slide. The resulting ROIs were saved in TIFF format. The statistical and geometric features (*n* = 32) of the 3D tumor were extracted using the Fiji 3D-ROI Manager plugin [45]. The texture features (*n* = 54 features) were computed by loading the TIFF ROIs (TIFF-stack library) into MATLAB R2018b using the texture toolbox [46,47]. Finally, the two feature sets were combined, resulting in 86 imaging traits for each LUAD patient.

### 4.3. Genomic Data Analysis

Gene and miRNA expression profiles were processed by normalization of raw read counts followed by differential expression analysis. We used the DESeq2 v. 1.12.4 R package [48] to identify differentially expressed genes (DEGs) and miRNAs (DEMs) between normal and tumor samples. Genes and miRNAs that exhibited at least a 2-fold change and a *p*-value cutoff of 0.05 were classified as DEGs and DEMs, respectively. *p*-values were adjusted using the Benjamini–Hochberg [49] procedure to limit the false discovery rate to 5%.

### 4.4. Enrichment Analysis of Differentially Expressed Genes and miRNAs

To compare the functional enrichment of the DEGs versus the DEMs, we used the GOSt tool of the gProfiler2 R package [27] with the correction method gSCS to identify significantly enriched (*p*-value < 0.05) GO biological processes.

To study the association between the transcriptomic functional level and the radiomic phenotypes, we used the gene set enrichment analysis (GSEA) implemented in the R package Piano [28]. For each combination of image features and a GO term, we performed GSEA to evaluate the Spearman rank correlation between the gene or miRNAs of the GO term and the image feature values. The *p*-values (<0.05) obtained from the GSEA were evaluated through 10,000 gene or miRNA set random permutations, and FDR-adjusted.

The summary statistic indicates the directionality of the association between the GO term and the image feature in the up or down direction, revealing positive and negative associations between the transcriptomic expression profiles and the image feature. 

In our further analysis, we restricted our evaluation by considering only GO terms with more than two image features significantly associated with GSEA for both gene and miRNA-based analysis. 

### 4.5. Visualization of the Radiogenomic Association Maps

Hierarchical clustering with Euclidean distance and the complete method (hclust R function) was used to derive a dendrogram of columns for visualization. T and N classification [4], smoking status, patient age, and mutation status of EGFR, KRAS, ALK, and TP53 were added. The heatmaps were visualized using the ComplexHeatmap R package [29].

### 4.6. Identification of Differentially Representative Features (DRF)

The fold change (FC) for each image feature between two patient groups was calculated and tested for significance using the unpaired statistical *t*-test. *p*-values were adjusted using the Benjamini–Hochberg [49] procedure to limit the false discovery rate to 5%.

### 4.7. Gene Regulatory Network Construction

The TFmiR2 web server [30] was utilized to construct the gene regulatory network (GRN) from the genes and miRNAs significantly associated with the examined GO terms with a *p*-value of less than 0.01. We contextualized the output network to lung cancer by selecting non-small cell lung carcinoma as the disease attribute. We also considered molecular interactions that were only supported by experimental evidence. The output networks were visualized by Cytoscape V.3.7.1 [50] highlighting edges/interactions that are lung-cancer and tissue-specific. All used methods and software packages are listed in Appendix A. 

## 5. Conclusions

We demonstrated a radiogenomics-based approach that deciphers the underlying regulatory machinery behind tumor imaging phenotypes by systematically correlating transcriptomic and image features in lung cancer patients. We have highlighted several biological processes significantly associated with tumor phenotypes (radiomic features) and unraveled the corresponding regulatory interactions with potential driver genes and miRNAs, providing better interpretability of radiologic phenotypes. This data-driven approach can be generalized to other cancer types and complex diseases, given the availability of related multi-omics datasets. Such an approach could be helpful in individualized medicine for detailed non-invasive diagnosis, treatment suggestions, drug susceptibility testing, and patient follow-up.

## Figures and Tables

**Figure 1 ijms-24-04947-f001:**
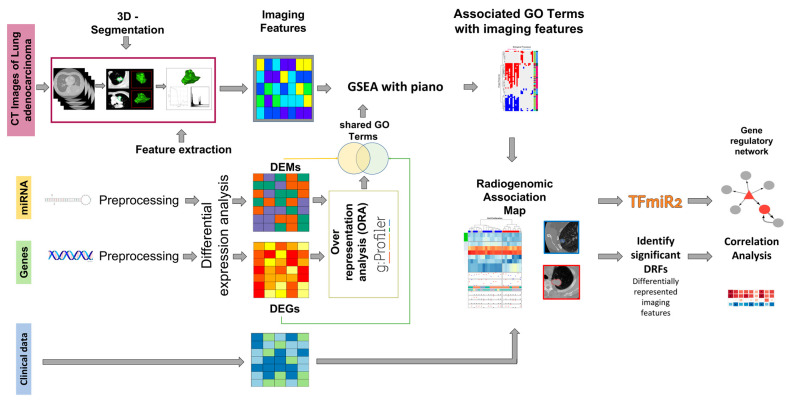
A schematic diagram of the integrative workflow. The diagram describes the image and transcriptomic data processing and the integration of the four different datasets into radiogenomic association maps. Starting from the preprocessing step, 515 mRNA samples, 513 miRNA samples, and 69 CT image series samples are reduced to the same patients by intersection, resulting in 22 patients for the integration steps. On the right, the further evaluation of the results by regulatory network construction and correlation analysis is outlined.

**Figure 2 ijms-24-04947-f002:**
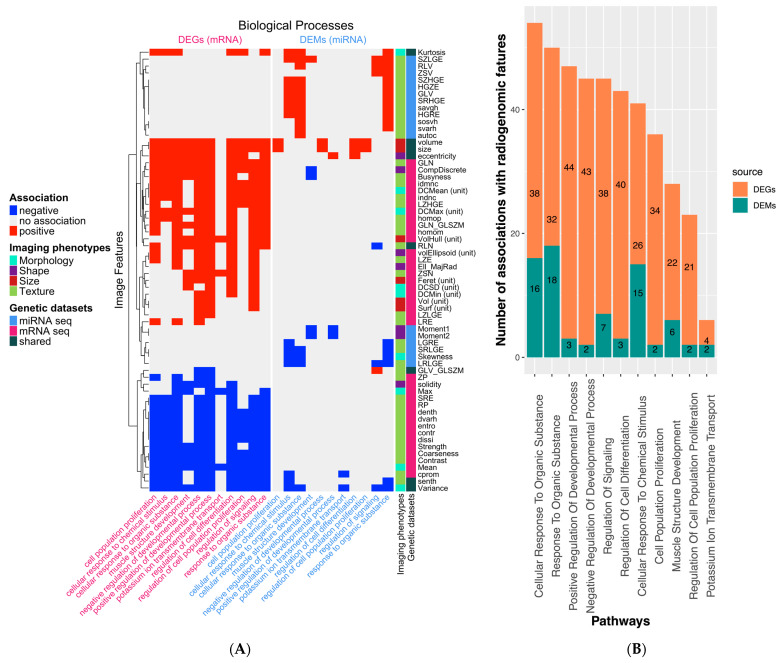
Visualization of the significant associations between the GO BP terms. (**A**) The Piano results of common “biological process” terms (*n* = 11) from the ORA of DEGs (pink-colored column) and DEMs (blue-colored column) are visualized (considering only image features with at least two negative or positive associations in the dataset). Image features in rows are clustered to reveal a different association pattern in the two transcriptomic profiles (DEGs and DEMs). The annotation column “Imaging phenotypes” reflects the assignment of image features to their group. The “Genetic dataset” legend characterizes an image feature if it was exclusively associated with the terms of gene expression (RNAseq, DEGs), miRNA expression (miRNAseq, DEMs), or within both datasets (shared). (**B**) The impact of the selected 11 GO terms is illustrated as a bar plot, where the number of significant associations between any image feature and the GO terms is split into DEGs and DEMs. The GO term with the most associations based on the GSEA with Piano is a cellular response to organic substances.

**Figure 3 ijms-24-04947-f003:**
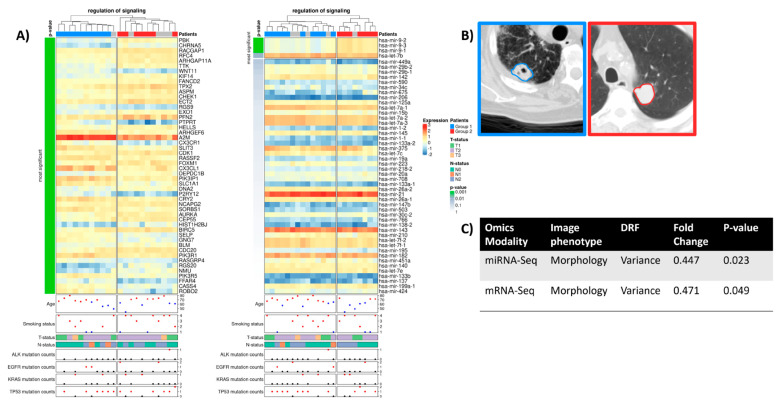
(**A**) Radiogenomic association map of the GO term “regulation of signaling”. The clustered rows are ordered by their *p*-value calculated using a *t*-test between the two groups. The first 50 genes/miRNAs are visualized. The gene expression-based RAM results from clustering 1001 genes, of which 322 have a *p* < 0.05 (32.2%). The miRNA-based RAM showcases four miRNAs with a *p* < 0.05 from 53 miRNAs used for clustering (7.5%). In addition, clinical information, tumor stage, and mutation frequency of common lung cancer driver genes are displayed. Smoking status was defined as follows: 1-lifelong non-smoker, 2-current smoker, 3-current reformed smoker for more than 15 years, and 4-current reformed smoker for less than 15 years. (**B**) Two exemplary contrast-enhanced CT images (level-600, window 1500) are presented on the right. (**C**) Table showing the identified DRFs and their significant association with the miRNA-Seq results of the GO term.

**Figure 4 ijms-24-04947-f004:**
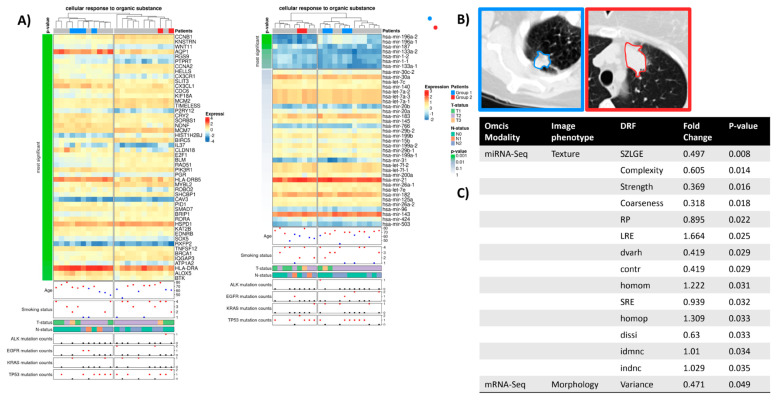
(**A**) Radiogenomic association map of the GO term “cellular response to organic substance”. The clustered rows are ordered by their *p*-value calculated using a *t*-test between the two patient clusters. The first 50 genes/miRNAs are visualized. The gene expression-based RAM results from clustering 794 genes, of which 246 have a *p* < 0.05 (31%). The miRNA-based RAM showcases 7 miRNAs with a *p* < 0.05 out of 39 miRNAs used for clustering (18%). In addition, clinical information, tumor stage, and mutation frequency of common lung cancer driver genes are displayed. Smoking status was defined as follows: 1-lifelong non-smoker, 2-current smoker, 3-current reformed smoker for more than 15 years, and 4-current reformed smoker for less than 15 years. (**B**) Two exemplary contrast-enhanced CT images (level-600, window 1500) are presented on the right. (**C**) Table showing the identified DRFs and their significant association with the miRNA-Seq results of the GO term.

**Figure 5 ijms-24-04947-f005:**
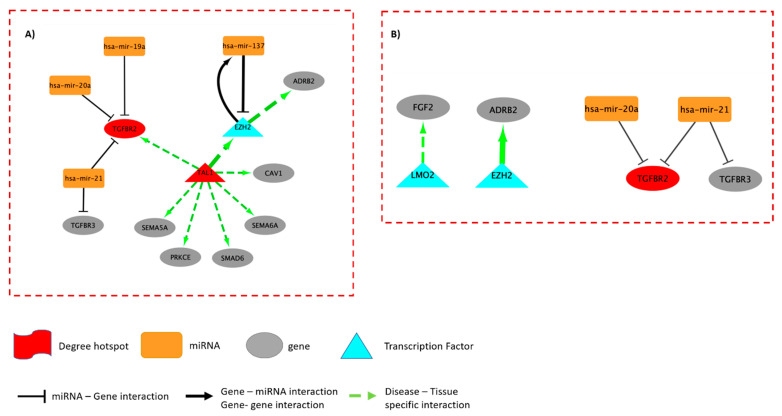
Gene regulatory network (Tfmir) of the GO term (**A**) “regulation of signaling” and GO term (**B**) “cellular response to organic substance”. TAL1 and TGFBR2 are regulatory hotspots based on the degree of centrality of the edges. Green arrows indicate tissue specificity for non-small cell lung carcinoma. Directly influenced target genes are grey, while orange represents miRNAs that act as inhibitors. TGF-beta receptor types 2 and 3 are both inhibited by hsa-mir-21.

**Table 1 ijms-24-04947-t001:** Demographic and clinical characteristics of the tumor patients. The available data are not always complete, as noted in the “No data” row. The “common” column refers to the integration step within the pipeline, where the preprocessed data are reduced by the intersection of patients between all datasets. The *t*-test statistic shows no significant difference between the groups based on clinical characteristics.

		mRNA Seq LUAD (*n* = 515)	mRNA Seq Normal (*n* = 59)	mRNA LUAD- vs. Normal	miRNA Seq LUAD (*n* = 513)	miRNA Seq Normal (*n* = 46)	miRNA LUAD vs. Normal	Common (*n* = 22)	Common vs. mRNA LUAD	Common vs. miRNA LUAD
				*t*-test statistic			*t*-test statistic		*t*-test statistic	*t*-test statistics
Cohort size	Clinical data	515	59		513	46		22		
	No data	0	0		0	0		0		
Age (years)	Median	66	66	−0.558 *p* = 0.578	66	67	−0.171 *p* = 0.865	67.5	0.294 *p* = 0.722	0.229 *p* = 0.821
	Min–Max	33–88	42–86		38–88	47–85		42–80		
	No data	19	0		19	0		0		
Gender	Female	276	34	/	274	26	/	14	/	/
	Male	239	25		239	20		8		
Pack-years smoked	Median	40	48	0.181 *p* = 0.857	40	40	0.290 *p* = 0.773	25	−0.574 *p* = 0.581	−0.591 *p* = 0.57
	Min–Max	0.15–154	5–94		0.15–154	2–124		10–120		
	No data	163	26		163	10		13		
Vital status at last follow-up	Alive	389	37	/	388	41	/	14	/	/
	Dead	126	22		125	5		8		
Last Follow-up	Median days	157	306	−1.73	157.5	182	−0.345	242	−1.18	−1.24
	No data	134	19	*p* = 0.091	133	4	*p* = 0.732	8	*p* = 0.251	*p* = 0.229
KRAS mutation	Tested and mutation found	23	3		23	2		3		
	Tested and no mutation found	36	3		36	2		8		
EGFR mutation	Tested and mutation found	23	1		23	-		2		
	Tested and no mutation found	57	4		57	3		10		

**Table 2 ijms-24-04947-t002:** Overview of the GO terms, retrieved from both the GSEA of the enriched GO terms from genes and miRNAs. Two patient clusters are constructed to compare the groups based on each GO term’s involved genes/miRNAs. We counted how many patients intersected in each group for the different transcriptomic source data. For the complete overlap of patients, the number of matching patients was summed and divided by the total number of patients (*n* = 22). For each GO term, there were also associated image features, which we called DRF (differentially represented feature), and we checked if these matched between the miRNA and gene-based GSEA outputs.

GO Terms	Patient Cluster 1	Patient Cluster 2	Overlap	Percent Overlap	DRFs Associated with DEGs	DRFs Associated with DEMs
**cell population proliferation**	11	1	12/22	55%	Variance	Variance
**regulation of cell population proliferation**	11	1	12/22	55%	Variance	RLN, GLN_GLSZM, ZSN
**positive regulation of developmental process**	7	1	8/22	36%	RLN, GLN_GLSZM, ZSN	RLV, volume, size, solidity, eccentricity
**potassium ion transmembrane transport**	9	5	14/22	64%	Ell_Flatness, Moment5, cprom	Moment4
**cellular response to chemical stimulus**	4	3	7/22	32%	Variance	LRLGE, SZLGE, Complexity
**response to organic substance**	5	2	7/22	32%	Variance	No DRF
**muscle structure development**	11	1	12/22	55%	RLN, GLN_GLSZM, ZSN	Moment4
**negative regulation of developmental process**	10	7	17/22	77%	Variance	Variance
**regulation of cell differentiation**	10	7	17/22	77%	Variance	Variance
**cellular response to organic substance**	4	2	6/22	27%	Variance	contr,dissi,homom, homop, dvarh,indnc,idmnc, SRE, LRE, RP, SZLGE, Coarseness, Complexity, Strength
**regulation of signaling**	10	7	17/22	77%	Variance	Variance

## Data Availability

The results published here are in whole or in part based upon data generated by the TCGA (https://www.cancer.gov/tcga) (accessed on 19 July 2019) and TCIA Research Networks: (https://www.cancerimagingarchive.net) (accessed on 29 January 2020).

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
