# Peer review of "Data-Driven Radiogenomic Approach for Deciphering Molecular Mechanisms Underlying Imaging Phenotypes in Lung Adenocarcinoma: A Pilot Study"

_ijms, 2023, doi:10.3390/ijms24054947_

Round 1
Reviewer 1 Report
After reading the entire article, I believe it can be acceptable with some modifications.
1. Clearify the training dataset. As well, describe all features of the tuning dataset.
2. The K-means cluster is not described well. Cite this paper, which describes K-means clustering in detail. https://doi.org/10.1016/j.health.2022.100112
3. The author should compare the outcome with some existing methods.
4. Applied models are not clear to readers; the author should describe properly how those models were applied to their dataset.
5. The overall literature on this paper is very poor.
6. Cite those works: https://doi.org/10.3390/cancers11091293, https://doi.org/10.1007/s11334-022-00523-w
Author Response
Reviewer 1
- Clarify the training dataset. As well, as describe all features of the tuning dataset.
- We thank the reviewer for this comment. Due to the small number of matched samples, we had to use the whole dataset as a training set to identify association patterns between the image phenotypes and genotypes. Further studies with larger cohorts of matched samples need to be conducted to confirm detected association patterns. We thoroughly expressed this issue in the last paragraph “Studies limitations”. As suggested by the reviewer we added a new sentence for clarification on page 14 line Additionally, we added supplementary table 4 listing the number of features for each data set and how they are acquired
- The K-means cluster is not described well. Cite this paper, which describes K-means clustering in detail. https://doi.org/10.1016/j.health.2022.100112
- We do agree with our reviewer on his comment. As suggested by the reviewer, we cited the K-means clustering algorithm in the introduction section referred to its applications in similar problems.
- The author should compare the outcome with some existing methods.
- As suggested by the reviewer, we have added another reference in the discussion section to other methods that either also use RAMs or work with correlation-based approaches on lung cancer cohorts.
- Applied models are not clear to readers; the author should describe properly how those models were applied to their dataset.
- We thank our reviewer for this comment. The first paragraph of the results section has been extensively modified to be clearer. We also added more information on the used tools and the applied models.
- The overall literature on this paper is very poor.
- As suggested, the literature has been enriched with recently published work, considering more machine learning-based approaches in radiomics, ongoing standardization efforts, new targetable mutations and options for immunotherapy
- Cite those works: https://doi.org/10.3390/cancers11091293, https://doi.org/10.1007/s11334-022-00523-w
- Thanks to the reviewer for pointing us to those two papers. We cited the first paper “A Radiogenomic Approach for Decoding Molecular Mechanisms Underlying Tumor Progression in Prostate Cancer” as it is closely related to the topic. The other link leads to a study named “An efficient Apriori algorithm for frequent pattern in human intoxication data” which we believe that is out of study scope and might cause confusion to the readers. Therefore, we prefer not to include the second one to avoid potential confusions.
Reviewer 2 Report
In this study, the authors proposed a radiogenomics study to understand the relationship between imaging features and genotype in lung adenocarcinoma. Although the idea seems interesting, some major concerns are raised as follows:
1. The authors first used bioinformatics approaches to characterize molecular mechanisms of lung adenocarcinoma, and then treat them as "ground truth" to analyze the radiomics features. In this way, these ground truths could not be verified as a gold standard before considering any analyses. Why did the authors not try some high-evidence genotypes in lung cancer such as EGFR, KRAS, etc.?
2. All results have been conducted on TCGA/TCIA data without further validation. As I knew, the number of overlap samples between TCGA and TCIA is a few. Thus, this amount of data cannot be enough to claim a significant finding.
3. The authors must have some external validation data.
4. It is unclear how the authors segmented the ROIs (tumors). Did they use some experience radiologists/physicians to help with it?
5. Statistical tests and p-values should be shown in Table 1.
6. The review of related work is not sufficiently thorough and not sufficiently specific.
7. The computational details are not mentioned. Which software, program languages, libraries, etc., were used to build this approach? Is it possible to publish code to use this architecture and reproduce the results?
8. More references related to radiomics-based lung cancer analysis should be added to attract a broader readership i.e., PMID: 34502160, PMID: 34298828.
9. The figures and tables quality needs to be improved.
10. Some technical aspects and essential insights of the proposed method are not described in detail.
11. Overall, English writing and presentation style should be improved.
Author Response
Reviewer 2
- The authors first used bioinformatics approaches to characterize molecular mechanisms of lung adenocarcinoma, and then treat them as "ground truth" to analyze the radiomics features. In this way, these ground truths could not be verified as a gold standard before considering any analyses. Why did the authors not try some high-evidence genotypes in lung cancer such as EGFR, KRAS, etc.?
- We thank our reviewer for this valuable comment. We would like to clarify that our approach characterizes the potential association links/patterns between the molecular features/correlates and imaging phenotypes based on enrichment analysis and unravelling the underlying interaction mechanisms. The ground truth was basically the disease signature of dysregulated genes/miRNAs that were employed here as the starting point of our radiogenomic approach.
- The Reviewer is also steering to an interesting suggestion of using high-evidence genotypes (EGFR and KRAS mutated cases) as a gold standard. Unfortunately, the final cohort size of matched samples between the 3 omics datasets did not cover enough patients for using their mutation status as a valid characteristic. Further studies with larger cohorts of matched samples with sufficient KRAS/EGFR mutated cases, need to be conducted to be able to drive potential association patterns. We thoroughly expressed this issue in the last paragraph “Study limitations”. We also added the number of EGFR and KRAS mutated cases in the different datasets in table 1.
- All results have been conducted on TCGA/TCIA data without further validation. As I knew, the number of overlap samples between TCGA and TCIA is a few. Thus, this amount of data cannot be enough to claim a significant finding.
- This is a very relevant comment from the reviewer, and it was basically the main challenge in this study, that is few numbers of matched samples between TCIA and TCGA repos. Therefore, we clearly stated that in the “study limitations” paragraph. Unfortunately, we had to rely on the existing publicly available datasets from the TCGA and TCIA projects in order to perform this pilot study. It is noteworthy to mention that the corresponding author is currently applying for a “Deutsche Forschungsgesellschaft” grant to generate larger cohort of matched multi-omics data sets, including epigenetics as an additional omics layer, to validate and give more insights on the identified patterns.
- The authors must have some external validation data.
- We relate to the above-mentioned point, that is due to the small number of the available matched samples in both TCIA and TCGA, we had to use the whole dataset as a training set to identify association patterns between the image phenotypes and genotypes. Additionally, we elaborated more on the necessity of external validation in the “study limitations” paragraph. To further stress this fact, we changed the title to: “A Data-Driven Radiogenomic Approach for Deciphering Molecular Mechanisms underlying Imaging Phenotypes in Lung Adenocarcinoma Patients, A pilot study”
- It is unclear how the authors segmented the ROIs (tumors). Did they use some experience radiologists/physicians to help with it?
- The ROI segmentation was performed by the second author who is a physician trained by expert radiologists in the field at the University Rostock medical center and he has got solid experience identifying tumor nodules. We added that the segmentation was done manually in the method section to be clear for other readers.
- Statistical tests and p-values should be shown in Table 1.
- As suggested by the reviewer, we have extended Table 1 to see if there is a significant difference between the groups with different characteristics.
- The review of related work is not sufficiently thorough and not sufficiently specific.
- We thank the reviewer for his comment. As suggested also by reviewer 1 too, we enriched the literature with more precise and up-to-date related article
- The computational details are not mentioned. Which software, program languages, libraries, etc., were used to build this approach? Is it possible to publish code to use this architecture and reproduce the results?
- Unfortunately, we cannot share the code of our approach publicly, because the code belongs to the Ph.D. work of the first two authors and will be released with the doctoral theses. However, to make the approach reusable, we listed the used software and libraries in supplementary table 5.
- More references related to radiomics-based lung cancer analysis should be added to attract a broader readership i.e., PMID: 34502160, PMID: 34298828. (https://www.mdpi.com/2072-6694/13/14/3616 and https://www.mdpi.com/1422-0067/22/17/9254)
- Done
- The figures and tables' quality needs to be improved.
- We provided the figures in a higher resolution and this point should be tackled after processing the manuscript through the journal submission system. In addition, we reformatted the tables to be more accessible to readers.
- Some technical aspects and essential insights of the proposed method are not described in detail.
- The reviewer is referring to the same point that was also raised by reviewer 1. Accordingly, we have elaborated on the used approach and methods. Please see changes in paragraph 1 in the results section as well as the added paragraph in the methods
- Overall, English writing and presentation style should be improved.
- We revised the manuscript with the help of a proficient English-speaking colleague. A final check will be also done before the processing of the manuscript.
Round 2
Reviewer 2 Report
My previous comments have been addressed.